# Microwave and Roasting Impact on Pumpkin Seed Oil and Its Application in Full-Fat Mayonnaise Formula

**DOI:** 10.3390/foods11182732

**Published:** 2022-09-06

**Authors:** Leila Rezig, Zina Harzalli, Karima Gharsallah, Nesrine Mahfoudhi, Moncef Chouaibi, Hatem Majdoub, Imen Oueslati

**Affiliations:** 1LIP-MB ‘Laboratory of Protein Engineering and Bioactive Molecules’, National Institute of Applied Sciences and Technology, University of Carthage, LR11ES26, Tunis 1080, Tunisia; 2High Institute of Food Industries, University of Carthage, 58 Alain Savary Street, Tunis 1003, Tunisia; 3Laboratoire de Biotechnologie de l’Olivier, Centre de Biotechnologie de Borj-Cedria, B.P. 901, Hammam-Lif 2050, Tunisia; 4Process Engineering Department, Higher Institute of Technological Studies of Zaghouan, General Direction of Technological Studies, Tunis 2098, Tunisia; 5Physics Laboratory of Soft Matter and Electromagnetic Modeling, Faculty of Science of Tunis, El Manar University, LR99ES16, Tunis 2092, Tunisia; 6Departement of Biotechnology, Faculty of Science and Technology of Sidi Bouzid, University of Kairouan, Sidi Bouzid 9100, Tunisia; 7Laboratory of Aromatic and Medicinal Plants, Biotechnology Center in Borj-Cedria Technopole, B.P. 901, Hammam-Lif 2050, Tunisia; 8Bio-Preservation and Valorization of Agricultural Products UR13-AGR 02, High Institute of Food Industries, University of Carthage, 58 Alain Savary Street, Tunis 1003, Tunisia; 9Laboratoire des Interfaces et des Matériaux Avancés, Faculté des Sciences de Monastir, Monastir 5000, Tunisia

**Keywords:** pumpkin seeds treatments, oil extraction, full-fat mayonnaise formulation, physical properties, antioxidant activity, storage

## Abstract

In this study, ‘Béjaoui’ *Cucurbita maxima* seeds variety were exposed to both microwave and roasting prior to oil cold press extraction. In addition, full-fat mayonnaise formula from untreated and treated pumpkin seed oils was prepared and assessed for their physical stabilities and bioactive properties in 28-day storage at 25 ± 1 °C. A mayonnaise sample prepared with unrefined sunflower seed oil served as a control. The results showed that the microwave pretreatment of seeds greatly enhanced the oxidative stability of the pumpkin seed oil, which increased from 3 h 46 min ± 10 min in the untreated sample to 4 h 32 min ± 14 min in the microwave cold press pumpkin seed oil. The sterol content increased from 4735 ± 236.75 mg/kg oil in the untreated cold press pumpkin seed oil to 5989 ± 299.45 mg/kg oil and 7156 ± 357.8 mg/kg in the microwave cold press pumpkin seed oil and the roasted cold press pumpkin seed oil, respectively. The mayonnaise prepared with microwave cold press pumpkin seed oil exhibited the lowest creaming index and was more stable to droplet growth when compared to the other mayonnaise samples. All mayonnaise samples prepared with pumpkin seed oils exhibited higher total phenolic contents and antioxidant activities during storage when compared to the mayonnaise sample prepared with unrefined sunflower seed oil. Among pumpkin seed oil mayonnaise samples, the highest values were, however, observed in the one prepared with microwave cold press pumpkin seed oil. Thanks to its high nutraceuticals, the latter could be confidently regarded as a natural fat substitute for commercial stable vegetable oils mayonnaise type emulsions.

## 1. Introduction

Pumpkin belongs to the Cucurbitaceae family, also known as cucurbits, a mid-sized plant genus that grows widely in moderate, tropical, and subtropical climates worldwide [1]. Pumpkin seeds are a good source of essential fatty acids and bioactive molecules such as β-carotene, α-tocopherol, vitamin B, lutein, phytosterols, and other minerals [2]. Moreover, among sterols, which lower the risk of cardiovascular mortality, the Δ7-sterols are specific to pumpkin seed oil and are supposed to confer to this oil a beneficial effect in the treatment and prophylaxis of the prostate gland and bladder disorders [3,4]. Thanks to such unique seed oil feature, pumpkin seed oil (PSO) is endowed with a strong antioxidant potential and interesting anti-inflammatory, antibacterial, and wound-healing properties [5,6]. PSO has also been recognized as an exceptional preventive against diabetes and carcinogenic diseases [7,8,9]. Pumpkin seeds are used in cooking mainly in southern Austria, Slovenia, and Hungary [10]. Moreover, roasted pumpkin seeds are a favorite snack in many African countries, including Tunisia. As PSO is abundant in oleic and linoleic acids, it is consumed as a salad or culinary oil in some countries. It is also used in margarine production [11,12,13]. Thanks to its richness in bioactive components, PSO is considered as preservative and functional ingredient [14,15].

It is of great importance to consider the effects of the various processing techniques used in oil extraction for human consumption. It is well known that the pressing is noticeably the common extraction method used for high oil content seeds. Unfortunately, this technique is quite ineffective due to the considerable amount of oil that remains in the crushed seeds. Given the growing demand for cold pressed oil and the lower extraction yield, thermal treatments of seeds prior to cold extraction such as microwave (MW) and roasting pretreatments are actually considered alternative available methods for high oil quality with coupled high availability of desirable nutraceuticals [16].

Electromagnetic (EM) waves with frequencies ranging from 0.3 GHz to 300 GHz are known as microwaves [15,16]. Microwaves heat foods volumetrically by converting electromagnetic (EM) energy to thermal energy within the food matrix, in contrast to conventional heating, which transfers heat from a high-temperature medium to a low-temperature medium [17]. The heat generation within materials is essentially due to molecular friction as a result of dipolar relaxation and ionic conduction [18]. As a result, microwave heating can significantly shorten the heating process, providing great convenience to consumers without inducing product quality degradation [17,18,19]. It is well known that microwave oven use has increased in homes despite some misunderstandings about the potential negative effects of microwave cooking on food nutrition and worries about microwave radiation to users. Today, food producers create a wide range of microwaveable frozen, chilled, and shelf-stable meals for retail markets to take advantage of microwave ovens’ benefits. In addition, microwave heating has been utilized in the food sector to thaw frozen meat and fish blocks, pre-cook bacon for fast food chains, and pasteurize pre-packaged foods, among other processing activities in order to minimize waste, increase throughputs, and improve food safety. In the food industry, adopting microwave technologies to replace some of the traditional heating techniques has additional benefits such as a cleaner work environment and a speeding up of the process [17]. In this context, Zhou et al. [19] studied the influence of microwave heating at two frequencies (2.45 and 5.85 GHz), both individually or in combination, in frying and post-frying on oil reduction in French fries. The latter showed that microwave frying reduced the frying time by 30–40%, with equivalent product quality attributes in terms of oil content, color, and texture, as compared to deep-oil frying.

Few investigations have been carried out on the impacts of pumpkin seed roasting on PSO’s quality and antioxidant characteristics. Ali et al. [20] reported microwave roasting as a quick processing technique that increases the oxidative and tocopherol stability, the antioxidant potential, and the levels of saturated fatty acids (SFAs) in PSOs. Furthermore, the roasting process at 175 °C for 15 min of pumpkin seeds has no meaningful impact on the total phenolic content of PSO. However, a marked decrease in tocopherol levels was observed leading to a decrease in its antioxidant activity [21]. Potočnik et al. [22] studied the effect of roasting temperatures ranging from 90 °C to 200 °C on PSO’s tocopherols and phenolic amounts. They pointed out a decrease in these bioactive compounds’ concentrations with increasing roasting temperature. Meanwhile, the antioxidant capacity of PSO increased with roasting up to 110 °C.

In terms of the effects of seed MW pretreatment on the oil’s quality, Delfan-Hosseini et al. [23] stressed that microwaved purslane seeds caused a rise in total phenolic content (TPC) and antioxidant activity accompanied with a significant enhancement in the oil’s oxidative stability. Improvement of the oil’s induction time by microwave pretreatment was consolidated by the findings of Uquiche et al. [24] on Chilean hazelnuts. As far as we are aware, there are little existing data on the influence of pumpkin seed MW pretreatment on PSO’s profile analysis and oxidative stability. In this regard, Yoshida et al. [25,26] showed that longer microwave processing caused an inferior percentage of linoleic acid and higher rates of oleic, palmitic, and stearic acids with superior levels of free fatty acids in *Cucurbita* (spp.) seed kernels. Moreover, in terms of tocopherols, more than 85% tocopherols remained after roasting for 20 min compared to untreated ones. Ali et al. [20] also reported that the more pumpkin seeds are exposed to microwave treatment, the higher oxidative indices of oil are observed. 

In line with the fact that increasing interest on health and well-being has shifted consumer preferences towards foods providing nutritional requirements and offering hedonistic pleasure, the incorporation of cold pressed PSO as a functional ingredient in some foodstuff formulations such as salad mayonnaise alone or mixed with refined sunflower seed oil, mayonnaise enriched with PSO-loaded microparticules, or reduced fat salad dressings, was the subject of recent investigations [27,28,29,30]. However, to the best of our knowledge, no data are available on the impact of roasting or microwave treatment of pumpkin seeds originating from North African countries and especially from Tunisia on oil quality and even in full-fat mayonnaise prepared with cold pressed PSO. Such a food product could be regarded as ‘functional’ even though the words ‘functional’ and ‘mayonnaise’ do not seem to correspond. Nevertheless, when considered from a different perspective, such a word combination could result in a ground-breaking and promising functional food. To make mayonnaise a healthful food, several strategies have been proposed: in addition to fat reduction, which has been the most widely used approach to date, a more recent method consists in enriching emulsions with healthy ingredients that can cater to people’s health-related needs (antioxidant, prebiotics, and probiotics).

Furthermore, from a food science viewpoint, functional components such as natural antioxidants are proven to enhance the oxidative stability of the emulsion and to permit the substitution of synthetic and controversial antioxidants. The aim of this research was to study the PSO’s physico-chemical and oxidative stability extracted from Tunisian ‘Béjaoui’ *Cucurbita maxima* seeds subjected to roasting and MW pretreatments prior to cold pressing, with respect to PSO extracted from untreated seeds and to investigate the physical properties, the antioxidant potential, and the bioactive compounds of a full-fat mayonnaise (≈80% oil) during 28 days storage at 25 ± 1 °C. For comparative purposes, a mayonnaise sample with unrefined sunflower seed oil was prepared as a control.

## 2. Material and Methods

All the reagents used in the assessment process were analytical or HPLC grade and were provided by Merck (Darmstadt, Germany).

### 2.1. Sampling

Pumpkin (*Cucurbita maxima*) seeds were brought from a plant nursery in December 2020 in the Chebika region (latitude 35°37′5.23″; longitude 10°2.15′38″; elevation 125 m), situated in west-central Tunisia. The seeds were first stored in stainless steel containers at 4 °C and 52% relative humidity, and only the undamaged ones were used. They were then rinsed to eliminate dirt, and lastly sun-dried until 11.11% moisture content (g/100 g wet basis) [31] was evaluated. Finally, until the seed pretreatments, seeds were divided into three equal portions of 15 kg each and stored in plastic bags away from light at 4 °C. The first portion was left untreated as a control.

### 2.2. Microwave and Roasting Pretreatments

For the second portion, 100 g of complete pumpkin seeds were uniformly layered in a Pyrex dish and treated in a domestic microwave oven (LG Electronics, model MS2042DW, 2450 MHZ, 1000 W). Microwave pretreatment was conducted according to Delfan-Hosseini et al. [23] at 700 W for 240 s. The real temperature of the pumpkin seed samples upon microwave treatment, determined with infrared thermometer (AR350, Smart Sensor, Shanghai, China), was of 96 ± 1 °C. 

As for the third portion, 2% water was added to the seeds and stirred fully to achieve a homogeneous blend, and then the mixture was roasted at 175 °C for 15 min according to Aktaș et al. [21] in a Binder FD series heating oven (BINDER GmbH, Tuttlingen, Germany). The seed layer thickness was 2.5 cm, and the temperature was monitored throughout the whole procedure. 

It is worth noting that the energy (E) required for microwave and roasting of pumpkin seed treatments were of 168 kJ and 3060 kJ, respectively (E = W Δt, where W corresponds to the power of the microwave oven (700 W) as well as that of the heating oven (3400 J) used, and Δt to the processing time expressed in seconds). 

For each seed pretreatment, three independent sets of experiments were performed under identical conditions in order to obtain homogeneous samples. Once the seed pretreatments were achieved, seeds were cooled at 25 ± 1 °C, wrapped in plastic bags, and reserved at 15 °C until the oil extraction process occurred.

### 2.3. Oil Extraction

Cold pressing was used to extract pumpkin seed oil using a Komet DD 85 G vegetable oil screw press (IBG Monforts Oekotec GmbH and Co. KG, Mönchengladbach, Germany) according to the procedure reported earlier by Rezig et al. [1]. Untreated cold press PSO, roasted cold press PSO, and microwave cold press PSO were named as UCPPSO, RCPPSO, and MWCPPSO, respectively.

### 2.4. Chemical and Physical Characterization of PSO

#### 2.4.1. Fatty Acid (FA) Profile Analysis

The fatty acid composition of PSO was analyzed by gas chromatography as fatty acid methyl esters (FAMEs) using the analytical methods described by the European Parliament and the European Council in EEC regulation 2568/91 [32]. 

#### 2.4.2. Tocopherol Composition

HPLC was used to determine the tocopherol composition of the seed oil using the NF EN ISO 9936 [33] according to Rezig et al. [7]. 

#### 2.4.3. Sterols Analysis (STs)

Sterols analysis was performed using the NF EN ISO 1228 method [34] according to Rezig et al. [1].

#### 2.4.4. 5-Hydroxymethylfurfural (HMF) Analysis

HMF was isolated from PSOs using a liquid-liquid extraction method and measured using an HPLC system (Agilent Infinity 1260, Agilent Technologies, Santa Clara, CA, USA) with a diode array detector (G1315D 1260 DAD VL, Agilent Technologies), quaternary gradient pump, and Agilent C18 column (250 × 4.6 mm, 5 µm, Agilent Technologies) according to the procedure reported in earlier works [35,36]. 

#### 2.4.5. FTIR-Analysis

FTIR analysis was employed in order to reveal the functional groups present in the PSO samples using an attenuated total reflectance mode (ATR-FTIR) spectrometer (Perkin Elmer 1600, Waltham, MA, USA). The FTIR spectra were documented in the absorption band mode in the range of 4000–400 cm^−1^ with a resolution of 4 cm^−1^ and 32 scans. PSO was analyzed without any previous treatment. 

#### 2.4.6. Oxidative Stability

The oxidative stability was determined in terms of the oxidative-induction time (OIT) using the 743 rancimat apparatus (Metrohm, Switzerland). The oil sample (3.5 g) was heated to 120 ± 1.6 °C in a reaction tube and subjected to an inflow of 20 L/h. OIT was automatically calculated by apparatus software.

### 2.5. Mayonnaise Preparation (O/W Emulsions)

According to Flamminii et al. [37], mayonnaise samples were prepared using the following formula from a reference recipe: 500 g oil (≈78%), five egg yolks (27 g ≈ 12%), vinegar (10 g ≈ 2%), salt (5 g ≈ 1%), lemon juice (37.5 g ≈ 7%), and sodium azide 0.05% (*w/w*). The samples were produced using an ultra-turrax T 25 homogenizer (IKA instruments, Staufen im Breisgau, Germany) following a two-stage standardized method: eggs, vinegar, salt, and lemon juice were preliminarily blended, and PSO was then gradually added under vigorous mixing at 20,000 rpm for 10 min, allowing complete oil incorporation. A mayonnaise sample prepared in the same conditions with unrefined sunflower oil commonly used in commercial mayonnaise formulation and furnished by the Med Oil Company society (Rades, Tunisia) served as control.

### 2.6. Emulsifying Properties

The emulsifying stability (ES) and activity (EA) were determined based on the method proposed by Alpizar-Reyes et al. [38]. 

### 2.7. Characterization and Stability of Mayonnaise during Storage

All analyses were performed on mayonnaise samples placed at 25 ± 1 °C at the production day (d_0_) and after 7, 14, 21, and 28 days of storage.

#### 2.7.1. Particle Size Measurement

By using laser diffraction analysis and a particle size analyzer (Mastersizer Hydro 3000, Malvern Instruments Ltd., Worcestershire, UK), the particle size distribution was determined according to Flamminii et al. [37]. Droplet size measurements were presented as the surface mean diameter (D_3,2_) and the volume mean diameter (D_4,3_).

#### 2.7.2. Creaming Index (CI)

The creaming index (CI) was evaluated as described by Alpizar-Reyes et al. [39].

#### 2.7.3. PH Measurement

The pH values were quantified in triplicate at 25 ± 1 °C with a pH meter (Mettler Toledo Co., Manchester, UK). In this study, the pH-meter was calibrated using standard pH 4.0 and pH 7.0 buffer solutions.

### 2.8. Bioactive Properties of Mayonnaise

#### 2.8.1. Evaluation of Antioxidant Activity (AA)

The AA of the mayonnaise samples were assessed by measurement of DPPH radical scavenging activity according to Romeo et al. [40].

#### 2.8.2. Total Phenolic Content (TPC)

The quantification of TPC was conducted on the oil separated from mayonnaise. The oil separation was performed according to Ozdemir et al. [41]. 

TPC of the obtained oils was measured colorimetrically using the Folin–Ciocalteu reagent [42] and reported as mg of gallic acid equivalents (GAE) per kg of oil. 

### 2.9. Oxidation Experiments in Mayonnaise

The oils were extracted from the mayonnaise samples according to Altunkaya et al. [43] with some adjustments as previously described in the Section 2.8.2. The peroxide value (PV) was determined according to AOCS Official Method Cd 8-53 [44].

### 2.10. Statistical Analyses

All experiments were carried out in triplicate. The results were expressed as mean values ± standard deviations (SD). Statistical analyses were performed on the data using the Statsoft statistical software package [45]. The Tukey’s test was applied after the one-way analysis of variance (ANOVA), and the differences between individual means and each used mean were deemed to be significant at *p* < 0.05.

## 3. Results and Discussion

### 3.1. PSO Characterization

#### 3.1.1. Fatty Acid Composition

As shown in Table 1, the main fatty acids in the UCPPSO were linoleic (45.2 ± 2.38%), oleic (34.45 ± 1.75%), palmitic (12.59 ± 1.64%), and stearic (6.35 ± 0.34%) acids. The average joined content of these four fatty acids was 98.89%, being slightly higher than that cited by Ali et al. [20] on *Cucurbita maxima* Duch. seed oil. Other fatty acids such as C14:0, C16:1, C18:3, C20:1, C22:0, and C24:0 were also present in low amounts. The fatty acid composition underwent no significant changes upon microwave treatment. In fact, the percentage of linoleic and oleic acids tended to fall slightly, while the percentage of arachidic, stearic, and palmitic acids raised slightly. This trend was most likely attributable to PUFA degradation, and it was consistent with what was observed by Yoshida et al. [26] on two Japanese pumpkin (*Cucurbita* spp.) cultivar seeds exposed to microwave treatment prior to oil extraction. In terms of roasting pretreatment, RCPPSO showed a minor drop in linoleic and oleic acids as well as a slight increase in palmitic and stearic acids, similar to MWCPPSO. Comparable results were observed by Aktaș et al. [21] on Turkish pumpkin (*Cucrubita pepo* L.) seeds. In fact, after roasting, the percentages of linoleic and oleic acids decreased from 42.72 ± 0.24% to 42.62 ± 0.21% and from 38.98 ± 0.0% to 38.45 ± 0.23%, respectively, in the ‘Nevşehir çerçevelisi’ variety. Furthermore, the percentages of stearic and palmitic acids increased from 7.22 ± 0.23% to 7.27 ± 0.13% and from 10.23 ± 0.22% to 10.9 ± 0.01%. According to Nederal et al. [46], such a feature is possibly a consequence of higher oxidation rate during the roasting process. Murkovic et al. [47] reported stability in oleic, palmitic, and stearic acids after pumpkin (*Cucurbita pepo* L.) roasting but a lowering in linoleic acid content which decreased from 54.6% to 54.2%. This small reduction was reported to be associated to oxidation products that were identified by headspace gas chromatography.

#### 3.1.2. Tocopherols

Table 1 lists the tocopherol contents of all PSO samples. Only α- and γ-tocopherols were identified with a predominance of γ-tocopherol accounting for 98.26%, 98.49%, and 98.86%, respectively, in UCPPSO, MWCPPSO, and RCPPSO. With respect to UCPPSO, pumpkin seed pretreatments had no significant effect on the γ-tocopherol contents which increased after MW pretreatment (383.08 ± 8.98 mg/kg) and decreased after roasting (349.54 ± 8.18 mg/kg). Meanwhile, a significant decrease was observed in the α-tocopherol content in RCPPSO (4.03 ± 0.16 mg/kg) accompanied with a non-significant decrease in MWCPPSO (5.87 ± 0.71 mg/kg). Our results regarding the decrease in γ-tocopherol content after roasting are in compliance with those cited by Aktaș et al. [21] on two Turkish varieties known as ‘Nevşehir Çerçevelisi’ and ‘Ürgüp Sivrisi’ which belong to the pumpkin species *Cucurbita pepo.* Such a feature was attributed to the limited roasting time applied during pumpkin seed pretreatment. The decrease in α-tocopherol content after roasting was not in accordance with the findings of Potočnik et al. [22] who noticed a significant increase of this tocopherol compound after roasting two Slovenic *Cucurbita pepo* L. cultivars (Gleisdorf and Rustikal) at a temperature range between 90 °C and 180 °C for 45 min. It is important to mention that the non-significant decrease in α-tocopherol observed in our study correlates, to some extent, to the findings of Yoshida et al. [26] on tocopherol contents in *Cucurbita* spp. seed oil who proved that the distribution patterns of tocopherol homologues (α-, β-, γ-, and δ-) were highly comparable, with a reduction of their amounts to less than 15% of the original levels after 20 min of MW pretreatments. The relative decrease in α-tocopherol content observed in the present study after applying MW pretreatment could be attributed, as reported by Azadmard-Damirchi et al. [48], to the decomposition of tocopherols by a relatively long period (4 min) of microwave pretreatment.

#### 3.1.3. Phytosterols

In the untreated cold press PSO sample (UCPPSO), fourteen phytosterols were identified, namely cholesterol, 24-methylenecholesterol, campesterol, campestanol, stigmasterol, Δ5-avenasterol, Δ5,23-stigmastadienol, Δ5,24-stigmastadienol, Δ7- campesterol, Δ7-stigmastenol, Δ7-avenasterol, clerosterol, sitosterol, and sitostanol (Table 1). Sitosterol and Δ5,23-stigmastadienol, which accounted for 1629.84 ± 81.44 mg/kg, were the predominant sterols, followed by Δ5,24-stigmastadienol (1395.87 ± 69.6 mg/kg) and Δ7-avenasterol (713.56 ± 35.51 mg/kg). Thus, sitosterol, Δ5,23-stigmastadienol, Δ5,24-stigmastadienol, and Δ7-avenasterol were the major sterols, and together they made up 78.97% of UCPPSO. It is important to point out that microwave treatment of pumpkin seeds prior to cold extraction had significant impact on the majority of the identified phytosterol compounds. In fact, with the exception of campesterol, campestanol, and stigmasterol, the amounts of which decreased after microwave pretreatment, all phytosterols exhibited an increase in their levels. With respect to UCPPSO, sitosterol + Δ5,23-stigmastadienol, Δ5,24-stigmastadienol, Δ7-avenasterol, Δ7-stigmastenol, and Δ7-campesterol suffered an increase of 16.88%, 29.02%, 39.15%, 32.03%, and 58.40%, respectively. These findings imply that microwave pretreatment causes disruption to the oilseed cell membrane, allowing for enhanced phytosterol release and increased quantities of phytosterols in extracted oil [16]. As far as we are aware, no previous studies were conducted on the impact of microwave pretreatment on the PSO phytosterol content. Otherwise, these results concur with previously published results conducted on rapeseeds proving the efficiency of MW pretreatment in increasing the phytosterol content by 15% in oil. With respect to UCPPSO, all phytosterols identified in RCPPSO suffered a significant increase in their amounts with a total rise of 51.13%. Similar results were observed in *Cucurbita pepo* L. seeds after roasting where the total sterol concentration increased from 1710 mg/g to 1930 mg/g. The rise in sterols throughout the roasting process was reported to changes in the seed meal as oil emerges from the seeds at the end of the roasting phase, altering the chemical behavior of the extraction process [47].

#### 3.1.4. HMF

In the current study, no HMF was detected in the UCPPSO and MWCPPSO samples (data not shown). Comparable findings were obtained by Suri et al. [49] in untreated and microwaved oils of Nigella seeds at 360 W (5 min) and 180 W (5 and 10 min). However, HMF levels in Nigella seed oil increased significantly as microwave power was raised from 540 to 720 W with the highest amount (2.08 mg/kg) recorded for those heated at 720 W for 10 min.

The HPLC analysis proved that pumpkin seed roasting at 175 °C for 15 min led to the production of HMF (0.41 ± 0.02 mg/kg). Such a finding was confirmed in a previous study conducted by Durmaz and Gökmen [35] in oil of pumpkin seeds roasted at 180°C for 30 min in which the amount of HMF was of about 4 mg/kg. A similar trend was found by Suri et al. [50] in oil of Nigella seeds subjected to infrared (160 °C, 10 min; 180 °C, 5 and 10 min) and dry air roasting (140 °C, 10 min; 160 °C, 5 and 10 min; 180 °C, 5 and 10 min). According to Durmaz and Gökmen [35], the content of HMF is affected by roasting time as well as compositional variables such as sugars, amino acids, oil, and moisture content of oilseeds.

#### 3.1.5. Evaluation by FTIR

The FTIR spectra, scanned using attenuated total reflectance mode, displays bands characteristic for the UCPPSO, MWCPPSO, and RCPPSO samples (Figure 1). Seven bands were observed in the UCPPSO at wavenumbers of 3675, 2973, 2902, 1394, 1231, 1066, and 892 cm^−1^. The two shoulder bands at 2973 cm^−1^ and 2902 cm^−1^ are associated respectively to asymmetric and symmetric stretching vibrations of C-H bonds of aliphatic CH_2_ triglyceride functional groups. Moreover, the small band of weak intensity at 1231 cm^−1^ and the sharp one observed at 1066 cm^−1^ are ascribed to the stretching vibration of C-O ester groups of the triglycerides [50]. Other studies describing the FTIR spectra of PSO registered sharp bands in the 2924, 2852, and 1745 cm^−1^ region revealing the prevalence of triglycerides and of other smaller intensity bands in the region of 3007, 1695, 1460, 1378, 1237, 1160, 1110, 1097, 950 and 850 cm^−1^, with some minor shifts in the bands to the left or to the right being recorded given the particularity of the oils composition [51]. It is worth mentioning that the small intensity band observed at 3675 cm^−1^, attributed to the telescopic vibration of hydroxyl [52], could be an indicator of the presence of humidity traces in the PSO sample. This band was present in the MWCPPSO sample with lower intensity than the one observed in the UCPPSO but was not detected in the RCPPSO, suggesting that the roasting process allows the complete dehydration of pumpkin seeds prior to cold extraction.

Otherwise, some changes occurred in the MWCPPSO and RCPPSO spectra when compared to the spectrum of the UCPPSO sample. In fact, in addition to the significant changes in intensities, it can be asserted that there was no variability in the band appearance with registered slow shifts to the right at the 2923 cm^−1^ and 2853 cm^−1^ regions. The increase in absorbance of these bands is due to the occurrence of chemical changes as a result of the oxidation process. This fact can be attributed to the high degree of oxidation of the samples [20]. Moreover, the MWCPPSO and RCPPSO samples exhibited four new sharp bands at approximately 1744 cm^−1^, 1465 cm^−1^, 1160 cm^−1^, and 724 cm^−1^ regions corresponding to saturated aldehyde (C = O) functional groups, bending vibrations of the CH_2_ and CH_3_, (C-O) stretching vibration, and overlapping of the methylene (= CH_2_) rocking vibration, respectively. The intensity of these bands was higher in RCPPCSO than in those observed in MWCPPSO.

This study’s findings revealed that the transmittance value of the bands was greatly affected by the pretreatment applied to pumpkin seeds prior to cold extraction. Furthermore, the roasting process seems to enhance the oxidation phenomenon with respect to MW pretreatment. 

#### 3.1.6. Oxidative Stability

The stability of UCPPSO was about 3 h 46 min ± 10 min (Table 1). Such a result is in agreement with that cited by Rezig et al. [1] on *Cucurbita pepo* var. ‘Essahli’ seed oil (3.74 ± 0.32 h). Microwave and roasting pretreatments improved the oxidative stability of the PSO samples. In fact, a significant increase in the oxidation induction time was observed in MWCPPSO and RCPPSO with respective values of 4 h 32 min ± 14 min and 3 h 50 min ± 20 min. The higher stability of oils extracted from microwave pretreated pumpkin seeds may stem from their high antioxidant content such as phytosterols (Table 1). Other findings reported by Azadmard-Damirchi et al. [48] reported that oil extracted from microwave-treated rapeseed shows a noticeably enhanced oxidative stability, most probably due to the increase of tocopherol content. Their findings are not in compliance with our results since microwave cold pressed pumpkin seed oil exhibited a non-significant increase in total tocopherol content when compared to UCPPSO. The increase of the oxidative stability upon the roasting process concurs with previous data cited by Vujasinovic et al. [53] on *Cucurbita pepo* L. seeds. In fact, after only 30 min of thermal treatment at 90 °C, the oil’s oxidative stability improved, and the induction period raised from 4.5 ± 0.05 h to 6.53 ± 0.09 h. With respect to UCPPSO, it can be said that, despite the appreciable content of unsaturated fatty acids (78.26%), pumpkin oil obtained from roasted seeds has a good oxidative stability. Apart from the richness of RCPPSO in tocopherols, and sterols (Table 1), the formation of Maillard reaction products such as HMF which amounted to 0.41 ± 0.02 mg/kg, known as potent natural antioxidants [54], could be responsible for the enhancement of the oxidative stability of the investigated pumpkin seed oil sample. 

### 3.2. Emulsifying Properties

The emulsifying ability of all mayonnaise samples was of 100% (Table 2). Such a result could be explained by the great efficiency of the emulsifying components in egg yolk (phospholipids, lipoproteins, livetin, and phosvitin) in retaining oil within the emulsion oil drops, allowing the absorption of surface-active oil molecules and the lowering of the surface tension [55]. Furthermore, the type of oil and particularly the fatty acid profile (amounts of tri-, di-, and mono-glycerides, fatty acid chain length, and the degree of saturation) were also reported to affect the surface tension at oil-water interfaces [56]. On the other hand, the highest emulsifying stability was recorded in MUCPPSO (98.18 ± 0.1%), followed in descending order by RCPPSO (96.36 ± 0.12%), MSFSO (89.66 ± 0.1%), and MWCPPSO (87.27 ± 0.13%) (Table 2). Significant differences in emulsifying stability between mayonnaise samples could be attributed to oil composition, fatty acid chain length, degree of saturation, ratio of mono-, di-, and triglycerides, squalene, and to various surface-active components such as cell membrane fragments, phospholipids, and vitamins [56,57]. 

### 3.3. Physical Stability of Mayonnaise during Storage

From Figure 2A, it can be seen that the creaming process started from the seventh day of storage and concerned the MUCPPSO, MMWCPPSO, and MRCPPSO samples. The highest CI values were observed in the mayonnaise sample prepared with untreated cold press PSO (MUCPPSO) which increased significantly (*p* < 0.05) from 8.3 ± 0.3% to 13.1 ± 0.67%. However, the lowest values were recorded in MMWCPPSO which raised from 1.25 ± 0.21% to 5.1 ± 0.41%. It is worth noting that the MSFSO (control sample) exhibited a creaming index value of 6.6 ± 0.26% only after 14 days of storage. This CI increased in a non-significant manner to reach a value of 7.2 ± 0.54% after 28 days of storage. With respect to MUCPPSO and MMWCPPSO, the CI values of the mayonnaise sample prepared with roasted cold press PSO were significantly lower than those observed in MUCPPSO and higher than those of MMWCPPSO. It seems that the microwaving process allows a better absorption of surface-active oil components at the interface enhancing the interfacial tension reduction and inhibiting the flocculation and the coalescence phenomena. The stability of the mayonnaise samples prepared with PSO over a week accords well with the findings of Anicescu et al. [58] who have demonstrated that an O/W emulsion prepared with PSO (50% *v/v*), distilled water (44% *v/v*), and lecithin as an emulsifier (6% *m/v*) maintains its stability for seven days after production. Otherwise, except for MUCPPSO which exhibited the highest CI value, a similar trend regarding the higher physical stability of the mayonnaise samples with PSO than that with sunflower oil during the last 14 days of storage was achieved by Nikolovski et al. [56]. These authors have in fact investigated the emulsions with PSO when compared to those prepared with sunflower oil.

In terms of oil droplet size (D_3,2_ values), the mayonnaise sample prepared with MWCPPSO exhibited the lowest particle size (0.24 ± 0.01 µm) at 0 d of storage when compared to the other mayonnaise samples (Figure 2B). At 7 days of storage, the D_3,2_ raised significantly to reach 1.39 ± 0.07 µm. However, it remained afterwards significantly stable reaching a value of 1.51 ± 0.12 µm. In contrast, MRCPPSO, showing an oil droplet size (1.82 ± 0.09 µm) significantly identical to MUCPPSO and MSFSO at 0 day of storage, presented a significant increase in the D_3,2_ value after 7 days of storage (3.37 ± 0.17 µm). Such a feature was maintained at 14 days of storage accompanied with a non-significant increase in oil droplet size value (3.42 ± 0.17 µm). During the last 14 days of storage, the oil droplet size doubled to reach the maximum value of 6.41 ± 0.15 µm.

Otherwise, it is worth mentioning that the oil droplet size of MUCPPSO increased in a non-significant manner between 0 day and 14 days of storage with respective values of 1.87 ± 0.09 µm and 2.18 ± 0.11 µm. A significant increase in D_3,2_ value was however observed after 7 days (2.53 ± 0.13 µm), after which it remained stable (2.52 ± 0.21 µm).

It is important to point out that, with respect to MSFSO and MRCPPSO, exhibiting the highest D_3,2_ values after 21 days and 28 days of storage, MUCPPSO and MWCPPSO displayed an oil droplet size not exceeding 3 µm.

The emulsifying stability was tested utilizing volume mean droplet size D_4,3_ as an indicator of the emulsion’s long-term stability [59]. Figure 2C depicts the D_4,3_ values of all mayonnaise samples during storage. It is worth mentioning that mayonnaise particle size was similar to that recorded by Worrasinchai et al. [60] (1–9 µm) and Laca et al. [61] (3–12 µm) but not to those documented by Liu et al. [62] (10–75 µm). These variances can be ascribed to the different components and preparation procedures used in the mayonnaises’ formulation. At 0 day, the MRCPPSO exhibited the highest D_4,3_ value (2.79 ± 0.14 µm) with respect to the other mayonnaise samples. During storage, the D_4,3_ value was remained stable reaching a maximum value of 3.18 ± 0.23 µm. It is essential to mention that the D_4,3_ values of MMWCPPSO featured the lowest values when compared to the other mayonnaise samples. In fact, at 0 day of storage, the D_4,3_ was of 0.63 ± 0.03 µm which increased significantly one week later to reach 1.53 ± 0.08 µm. No significant changes occurred subsequently during storage. From a statistical point of view, the evolution of the D_4,3_ in MUCPPSO and MSFSO was similar during storage. In fact, at 0 day, the respective values were of 2.31 ± 0.21 µm and 2.15 ± 0.11 µm. The latter increased in a significant manner after 14 and 21 days of storage to reach the maximum values of 4.3 ± 0.22 µm and 3.8 ± 0.14 µm. According to Laca et al. [61], the growth of the mayonnaise particle size after storage is attributable to the coalescence of oil droplets, which occurs after the droplets have been in contact for a long time in these close packed emulsions with high oil concentrations.

Furthermore, it is worth mentioning that the remarkable increase of the D_4,3_ values in MSFSO and MUCPPSO during the last 14 days of storage corroborates well with the CI evolution exhibiting the highest values, particularly in MUCPPSO (13.1 ± 0.89%).

In terms of pH, the initial pH values were between 3.14 and 3.39 at day 0 with no significant difference between the MUCPPSO, MMWCPPSO, and MSFSO samples and the MRCPPSO (Figure 2D). The highest value was obtained in the MRCPPSO (3.39 ± 0.01) while the lowest value was found in the MUCPPSO and MMWCPPSO samples (3.14 ± 0.00). The obtained values were in the same range as the one cited by Laca et al. [61] on a mayonnaise sample prepared with sunflower oil (≈86%), egg yolk (≈12%), and vinegar (≈11%). 

During the storage period, the pH values raised significantly with variances between stored mayonnaises. It is worth noting that the MMWCPPSO displayed the lowest pH values during storage with no significant difference between day 0 and 7 and during the last 14 days of storage.

After 28 days, the highest and lowest pH values in the stored MUCPPSO and MMWCPPSO mayonnaise samples were 4.01 ± 0.01 and 3.34 ± 0.01, respectively. The lowest pH values of MMWCPPSO and the highest ones of MUCPPSO during storage with respect to the other mayonnaise samples correlate with the lowest creaming index value found in the MMWCPPSO and the highest one observed in the MUCPPSO. A similar feature was observed by Wang et al. [63] who assessed the possibility of developing oil-in-water emulsions through unmodified natural egg-yolk granules (EYGs) at various pH levels, ranging from 2 to 9. The average size of the emulsions prepared with 1% EYGs was increased with the increase in pH. At pH 3.0 and 3.7 under the isoelectric point of the EYGs (pI = 4), the size distribution became uniform and narrow. Typically, at the isoelectric point of egg-yolk granules, the emulsion contained an extremely uniform size of about 50 μm. The size distribution was continuously increased at pH > pI to over 200 μm with the increase in pH, indicating that EYGs had a poor ability to stabilize emulsions at this pH.

### 3.4. TPC and Antioxidant Activity 

At day 0, the highest and lowest amounts of total polyphenol content were observed respectively in the lipid phases of MMWCPPSO (160 ± 0.74 mg EAG/kg) and MSFSO (5.11 ± 0.08 mg EAG/kg) (Figure 3A). After 7 days, and at the exception of MSFSO, the TPC decreased in a significant manner in all mayonnaise samples (*p* < 0.05) to reach 92.67 ± 0.72 mg EAG/kg, 57.34 ± 0.36 mg EAG/kg, and 53.44 ± 0.22 mg EAG/kg in MMWCPPSO, MRCPPSO, and MUCPPSO, respectively. Such a decrease in TPC was maintained subsequently during storage. After 28 days, a significant difference was observed between all mayonnaise samples. The highest and lowest contents were found respectively in MMWCPPSO (49.49 ± 1.22 mg EAG/kg) and in MSFSO (2.76 ± 0.01 mg EAG/kg). Meanwhile, no significant difference was observed between MRCPPSO (9.57 ± 0.26 mg EAG/kg) and MUCPPSO (10.25 ± 0.49 mg EAG/kg). As far as we know, no research study pertaining to describing TPC in the lipid phase of mayonnaises prepared with Cucurbitaceae seed oil has been published in the literature. The decrease in TPC could be explained by the transfer of phenolic compounds from the lipophilic to the hydrophilic phase thanks to their solubility. Such a finding correlates well with those of Romeo et al. [40] on mayonnaise samples enriched with different concentrations of olive leaf phenolic commercial extract and stored at 30 °C for 30 days. Further studies should be conducted to identify the nature of the phenolic compounds concerned by such a transfer from the lipophilic to the hydrophilic phases in all mayonnaise samples during storage.

The antioxidant activity using DPPH is represented in Figure 3B. The findings indicated that antioxidant activity at day 0 was the lowest in the MSFSO, reaching 12.11 ± 0.07%, while the MMWCPPSO exhibited the highest activity which was 54.71 ± 0.28%. With respect to MSFSO, the high antioxidant activity featured by MUCPPSO, MMWCPPSO, and MRCPPSO could be attributed to their richness in bioactive compounds. During the storage phase, the results revealed a significant decrease of antioxidant activity in all mayonnaise samples. Such a result indicates that some antioxidant compounds had escaped the emulsion. After 28 days, it was found that the lowest and highest active antioxidant activities were recorded in MSFSO (6.2 ± 0.08%) and MMWCPPSO (15.18 ± 0.06%). These results prove that there is a release of the compounds responsible for the antioxidant activity from the emulsion during the storage phase. Such a hypothesis might be consolidated by the decrease in TPC in the lipophilic phase of all emulsions tested since both experiments (TPC and AA) were conducted using a mixture of MeOH:H_2_O (60:40) or methanol alone in the extraction procedures (Figure 3A). Our findings are in compliance with those cited by Romeo et al. [40]. In fact, a decrease in total antioxidant activity assessed by DPPH was noticed in mayonnaise samples enriched with olive leaf phenolic commercial extract during storage. The reduction of % of inhibition was reported to be probably due to the positive effect of antioxidants on the mayonnaise samples.

### 3.5. Changes in the PV

Lipid oxidation is enhanced by the reaction at the surface of droplets of the O/W emulsion. Peroxide values are employed to determine the initial products (peroxide and hydroperoxide) of lipid oxidation. After mayonnaise preparation (day 0), the highest peroxide value was observed in MRCPPSO (7.5 ± 0.45 meqO_2_/kg), followed by MMWCPPSO (4.9 ± 0.67 meqO_2_/kg). No significant difference was however observed between PV of MUCPPSO and MSFSO, both exhibiting the lowest values. The peroxide values of all mayonnaise samples rose significantly afterwards (*p* < 0.05) with the prolongation of storage and attained the highest values after 28 days (Figure 4). The higher peroxide value observed in MSFSO (23.66 ± 0.13 meqO_2_/kg) with respect to the other mayonnaise samples could be attributed to the richness of SFSO in PUFA (58.37 ± 0.65%) (data not shown) when compared to UCPPSO (44.13 ± 2.21%), MWCPPSO (44.44 ± 2.22%), and RCPPSO (47.7 ± 2.38%) (Table 1). In fact, mayonnaise’s low pH causes the iron bridges between egg yolk proteins to collapse, which releases iron that can contribute to lipid oxidation by encouraging interactions with unsaturated lipids, thus producing lipid radicals or causing peroxide degradation [64,65]. A similar trend regarding the evolution of the peroxide value in mayonnaise samples during storage was cited by Li et al. [66] and Raikos et al. [67] once purple corn extract and beetroot were added, respectively. These authors claimed that these materials rich in phenolic compounds served as antioxidants, preventing or delaying lipid oxidation in mayonnaise. This finding is well consolidated in our present study. In fact, independently of the mayonnaise sample analyzed, the decrease in TPC and DPPH is attributed to oxidation, as evidenced by the sharp increase in PV values.

With respect to MMWCPPSO and MRCPPSO, the lower peroxide value observed in MUCPPSO despite its richness in TPC could however be attributed to other bioactive compounds preventing the oxidative process. 

## 4. Conclusions

The main purpose of this study was to examine the effects of microwave and roasting pretreatments on the quality and the oxidative stability of a Tunisian ‘Béjaoui’ cold pressed pumpkin (*Cucurbita maxima*) seed oil. In addition, we attempted to evaluate the potential of the obtained functional oils in ensuring a stable, full-fat mayonnaise sample during storage at 25 ± 1 °C as a substitute of commercial vegetable oils. The results revealed that the fatty acid and the tocopherol compositions underwent no significant modifications upon microwave and roasting treatments. It is important to point out that the heat pretreatments of pumpkin seeds enhanced the total sterol contents in MWCPPSO (5989 ± 299.45 mg/kg oil) and RCPPSO (7156 ± 357.8 mg/kg oil). The effect of microwave treatment on pumpkin seeds was more pronounced compared to roasting in improving the oxidative stability of PSO. 

As regards the physical stability of mayonnaise samples prepared with SFSO and PSOs, the mayonnaise sample prepared with MWCPPSO exhibited the lowest creaming index (CI) and was more stable to droplet growth during storage. Moreover, MMWCPPSO featured the highest amount of polyphenol content with an interesting antioxidant potential with respect to the other samples during storage. 

In the light of the obtained results and thanks to the lower energy consumption of microwaves compared to that of the roasting process, microwaved cold pressed PSO could be considerably valorized as a natural fat substitute of commercial vegetable oils in full-fat mayonnaise type emulsions.

## Figures and Tables

**Figure 1 foods-11-02732-f001:**
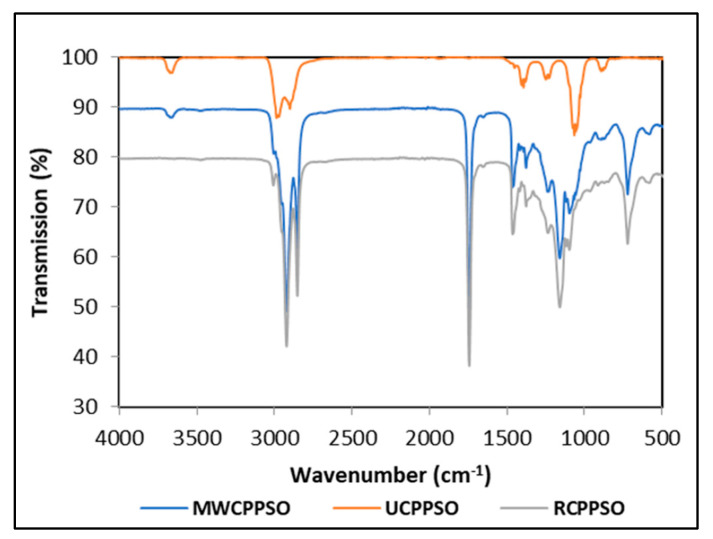
The FTIR spectra of PSOs.

**Figure 2 foods-11-02732-f002:**
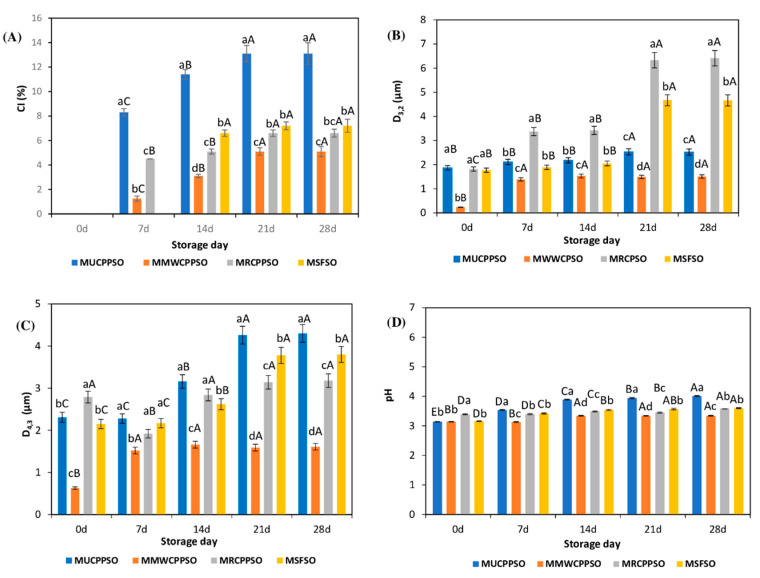
Evolution of the CI (%) (**A**), D_3,2_ (µm) (**B**), D_4,3_ (µm) (**C**), and pH values (**D**) of the mayonnaises during storage. Values are means ± standard deviations (SD) of three independent determinations. Different lowercase letters represent significant differences between the mayonnaises (*p* < 0.05). Different uppercase letters represent significant differences between storage times of each mayonnaise (*p* < 0.05).

**Figure 3 foods-11-02732-f003:**
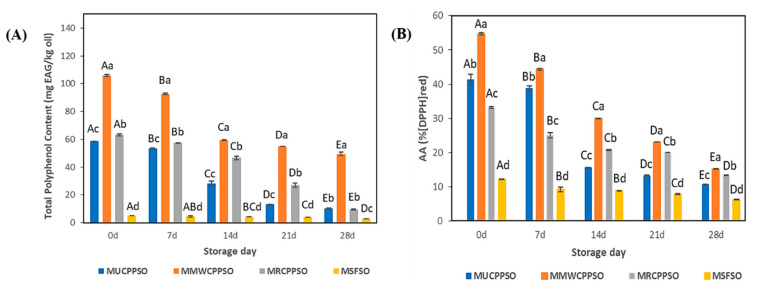
Evolution of total polyphenol content (mg EAG/kg oil) (**A**), and antioxidant activity (% (DPPH)red) (**B**) of the mayonnaises during storage. Values are means ± standard deviations (SD) of three independent determinations. Different lowercase letters represent significant differences between the mayonnaises (*p* < 0.05). Different uppercase letters represent significant differences between storage times of each mayonnaise (*p* < 0.05).

**Figure 4 foods-11-02732-f004:**
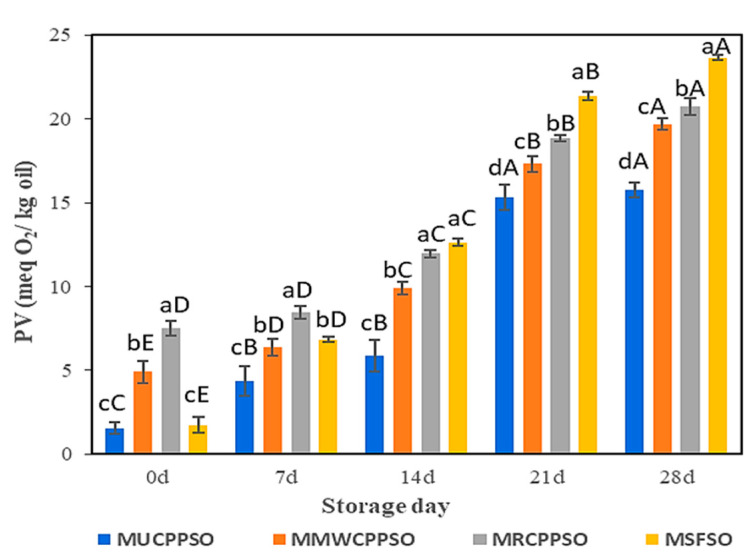
Evolution of the peroxide value of the mayonnaises during storage. Values are means ± standard deviations (SD) of three independent determinations. Different lowercase letters represent significant differences between the mayonnaises (*p* < 0.05). Different uppercase letters represent significant differences between storage times of each mayonnaise (*p* < 0.05).

**Table 1 foods-11-02732-t001:** Fatty acid (%), sterol (mg/kg oil), and tocopherol compositions (mg/kg oil), and oxidative stability of cold pressed pumpkin seed oils (a).

	Untreated Cold Press PSO (UCPPSO)	Microwave Cold Press PSO (MWCPPSO)	Roasted Cold Press PSO (RCPPSO)
**Fatty acids**	**Composition**		
**Myristic (C14:0)**	tr.	tr.	tr.
**Palmitic (C16:0)**	13.59 ± 0.64 a	14.05 ± 0.7 a	14.33 ± 0.65 a
**Palmitoleic (C16:1)**	tr.	tr.	tr.
**Stearic (C18:0)**	6.35 ± 0.34 a	6.54 ± 0.36 a	6.85 ± 0.34 a
**Oleic (C18:1)**	34.45 ± 1.75 a	34.37 ± 1.73 a	33.95 ± 1.48 a
**Linoleic (C18:2)**	45.20 ± 2.38 a	44.44 ± 2.22 a	44.29 ±2.19 a
**Linolenic (C18:3)**	tr.	tr.	tr.
**Arachidic (C20:0)**	0.38 ± 0.13 a	0.46 ± 0.09 a	0.35 ± 0.12 a
**Eicosenoic (C20:1)**	tr.	tr.	tr.
**Behenic (C22:0)**	tr.	tr.	tr.
**Lignoceric (C24:0)**	tr.	tr.	tr.
**SAFA**	20.33 ± 1.04 a	21.06 ± 1.01 a	21.54 ± 1.02 a
**MUFA**	34.46 ± 1.77 a	34.38 ± 1.68 a	33.96 ± 1.45 a
**PUFA**	45.21 ± 2.28 a	44.45 ± 2.12 a	44.30 ± 2.11 a
**Sterol**	**Composition**		
**Cholesterol**	9.47 ± 0.47 b	11.97 ± 0.60 a	12.16 ± 0.71 a
**24-methylenecholesterol**	26.51 ± 1.42 b	28.74 ± 1.19 b	32.20 ± 1.43 a
**Campesterol**	53.03 ± 2.36 b	44.31 ± 2.39 c	62.25 ± 2.86 a
**Campestanol**	20.83 ± 0.94 a	17.96 ± 5.39 a	25.04 ± 1.43 a
**Stigmasterol**	115.53 ± 5.68 a	107.80 ± 80 a	127.37 ± 6.44 a
**Δ5-Avenasterol**	37.88 ± 1.89 b	38.32 ± 1.79 b	46.51 ± 2.14 a
**Δ5,24-Stigmastadienol**	1395.87 ± 69.60 c	1801 ± 89.83 b	2090.98 ± 104.47 a
**Δ7-Campesterol**	148.20 ± 7.10 c	234.76 ± 11.38 b	269.78 ± 13.59 a
**Δ7-Avenasterol**	713.56 ± 35.51 c	992.97 ± 49.70 b	1162.13 ± 57.96 a
**Δ7-Stigmastenol**	425.67 ± 21.30 c	561.16 ± 28.14 b	646.18 ± 32.20 a
**Clerosterol**	60.13 ± 2.84 b	67.67 ± 3.00 a,b	71.56 ± 3.57 a
**Sitosterol + Δ5,23-Stigmastadienol**	1629.84 ± 81.44 c	1905.07 ± 95.22 b	2400.89 ± 119.50 a
**Sitostanol**	98.48 ± 4.73 c	177.27 ± 8.98 b	208.95 ± 10.73 a
**Total**	4735 ± 236.75 c	5989 ± 299.45 b	7156 ± 357.8 a
**Tocopherol**	**Composition**		
**α-Tocopherol**	6.71 ± 0.8 a	5.87 ± 0.7 a	4.03 ± 0.16 b
**γ-Tocopherol**	380.49 ± 38.63 a	383.08 ± 8.98 a	349.54 ± 8.18 a
**Total**	387.2 ± 39.43 a	388.95 ± 9.69 a	353.57 ± 8.34 a
**Oxidative Stability (h)**	3 h 46 min ± 10 min b	4 h 32 min ± 14 min a	3 h 50 min ± 20 min a,b

(a): Values with different letters in the same raw are significantly different; PSO: pumpkin seed oil; SAFA: saturated fatty acids; MUFA: monounsaturated fatty acids; PUFA: polyunsaturated fatty acids; tr.: trace amounts (less than 0.2%). Values are means ± standard deviations (SD) of three determinations.

**Table 2 foods-11-02732-t002:** Emulsifying properties of mayonnaise samples (a).

Samples (b)	MUCPPSO	MMWCPPSP	MRCPPSO	MSFSO
**Emulsifying ability (%)**	100 ± 0.00	100 ± 0.00	100 ± 0.00	100 ± 0.00
**Emulsifying stability (%)**	98.18 ± 0.1 a	87.27 ± 0.13 d	96.36± 0.12 b	89.66 ± 0.1 c

(a): Values with different letters are significantly different. Values are means ± standard deviations (SD) of three determinations; (b): MUCPPSO: Mayonnaise prepared with untreated cold press pumpkin seed oil; MMWCPPSO: mayonnaise prepared with microwave cold press pumpkin seed oil; MRCPPSO: mayonnaise prepared with roasted cold press pumpkin seed oil; MSFSO: mayonnaise prepared with unrefined sunflower seed oil.

## Data Availability

Data is contained within the article.

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
