# Peer review of "Microwave and Roasting Impact on Pumpkin Seed Oil and Its Application in Full-Fat Mayonnaise Formula"

_foods, 2022, doi:10.3390/foods11182732_

Round 1

Reviewer 1 Report

Keywords: Most of the keywords are similar to the titles, so they should be substituted to enhance the search index compatibility.

Introduction: It is divided into many short paragraphs, which affects the readability and flow, so this should be written in fewer and interconnected paragraphs.

The authors failed to express the novelty of this study and how this study is different from several similar studies conducted earlier:

https://link.springer.com/article/10.1007/s11746-012-2076-0

https://www.tandfonline.com/doi/full/10.1080/10942912.2016.1244544

https://onlinelibrary.wiley.com/doi/full/10.1002/ejlt.201100158

https://www.sciencedirect.com/science/article/pii/S0040603118304465

https://www.ocl-journal.org/articles/ocl/full_html/2018/03/ocl170032/ocl170032.html

Furthermore, in my opinion, preparing mayonnaise with some other oil doesn’t qualify as a healthy and novel food. Also, to prepare mayonnaise, commonly used oils such as soybean and sunflower also contain natural antioxidants, so this is not something new or different.

Line 171: "utomatically" ??

Line 221: "UCPPSO" ?? not explained anywhere in the manuscript.

Line 231: RCPPSO??

Line 233: MWCPPSO?? Provide the full form for these abbreviations.

Table 1: Table 1 shows a significant increase in the cholesterol content and a decrease in alpha-tocopherol post microwave and roasting in PSO. These results clearly contradict the authors' goal of making healthy mayonnaise. Please clarify ??

If possible for this study or for the future, I would suggest presenting the FTIR results as Absorbance vs Wavenumber as it provides a better understanding.

Line 357-359: The authors point out the increase in the tocopherol content as being a reason behind improved oxidative stability, but the results show no significant increase in the tocopherol content after microwave and roasting. Where there is a significant decrease in the alpha-tocopherol content after roasting. I would suggest reviewing the findings and just not stating the reasons as stated in other studies. Authors need to critically analyze and explain their findings.

The values of OIT are not presented anywhere in the manuscript, either in a table or graph, but just stated, which provides only partial and biased results.

Authors need to critically and clearly explain the results of oxidative stability.

Emulsifying ability may also be affected by the oil composition/fatty acid profile. Authors may refer to this article: https://www.sciencedirect.com/science/article/pii/S0927775719307769?via%3Dihub to better explain and understand their results.

From most literature, it can be pointed out that the most stable and safe mayonnaise should have a pH of 4.00–4.10. However, the pH values in this study were much lower, which may have a significant effect on stability rather than storage. As pH has a major effect on the droplet size. Please clarify?

Significant decrease in the TPC and DPPH may be attributed to the oxidation of mayonnaise over a storage period. Antioxidants are consumed during oxidation and till the antioxidants are present, it is referred to as the inhibited phase. Once all the antioxidants are consumed, the oxidation is out of control and is referred to as the uninhibited phase. The decrease in TPC and DPPH is attributed to oxidation, as evidenced by the sharp increase in PV values. Authors can correlate the findings of TPC, DPPH and PV to better understand the changes.

Reviewer 2 Report

The manuscript entitled " Microwave and Roasting Impact on Pumpkin Seed Oil and its Application in Full-fat Mayonnaise Formula” studied the influence of microwave and roasting pre-treatments on the quality of oils extracted from pumpkin seeds. Moreover, the storage stability of a mayonnaise sample containing the obtained oil was evaluated. The manuscript was well written, and the data reported in the study should be of interest to the research community and the food industry.  However, some issues in the manuscript should be addressed.

The following are my specific comments and suggestions.

Abstract

Line 30, oxidative stability of the seed oil was 4.53 hrs. It was confusing.

Line 31, increased from ? to 5989 and 7156 mg/kg. Suggest including data of control samples.

Line 34, “was more stable to droplet growth”. Compared to control or the roasting method?

Introduction.

Microwave heating prior to the extraction was the main treatment in this study. However, microwave heating and its application in food industry, especially on high-fat food products were not introduced properly. Please see the following recently published papers.

1.       Zhou, X., Zhang, S., Tang, Z., Tang, J., Takhar, P.S. 2022. Microwave frying and post-frying of French fries. Food Research International. 159:111663.

2.       Tang, J. 2015. Unlocking potentials of microwaves for food safety and quality. Journal of Food Science. 80(8) E1776-1793.

3.       Zhou, X. & Wang, S., 2019. Recent developments in radio frequency drying of food and agricultural products: A review, Drying Technology, 37:3, 271-286, DOI: 10.1080/07373937.2018.1452255

Materials and Method.

Many measurement methods were not described clearly, so with little information in this manuscript readers are not able to duplicate what authors have done. Even though authors refer to some literature, a brief description of how authors conducted experiments is recommended. In particular, if there is any difference between their own methods and methods used in the literature, authors need to explain why. Please revise the corresponding parts thoroughly.

Line 64, suggest change “meal” to “crushed seeds”.

Line 121, please indicate storage conditions (temperature, relative humidity, etc.)

Line 123, suggest change “fresh weight” to “wet basis (w.b.)” to describe moisture content

Line 128, “and domestically microwaved”. Suggest change it to “treated in a domestic microwave oven”.

Line 129, please delete “irradiation”. It is confusing for readers whether microwave is radiation or not. Microwaves at 2450MHz only have heating effect on foods, and no non-thermal effect (radiation) has not been verified.  Please see, Tang, J. 2015. Unlocking potentials of microwaves for food safety and quality. Journal of Food Science. 80(8) E1776-1793.

Line 138, please indicate room temperature. It varies significantly depending on the locations on Earth.

Line 188, what is the reason for 28-day storage? What is normal shelf-life of mayonnaise samples?

Line 192, change “are” to “were”.

Line 198, Calibration procedure of pH meter should be included.

Line 241, reference to support this statement?

Table 1, “Values with different letters in the same raw are significantly different; PSO: 243 pumpkin seed oil; SAFA: saturated fatty acids; MUFA: monounsaturated fatty acids; PUFA: poly-244 unsaturated fatty acids; tr.: trace amounts (less than 0.2 %). Values are means ± Standard Deviations 245 (SD) of three determinations.” Suggest move it to the footnote below the table. Similarly, please address this issue in other tables.

References, please use consistent format and double check references [2], [25], [44] etc. As mentioned earlier, recently published literature on microwave heating and its application should be added.

Reviewer 3 Report

Increasing of oil extraction efficiency helps to achieve better economy of technologies. Although, pre-treatments have effect on oil quality. Microwave treatments/pre-treatments are considered as promising processes for these purposes.  Manuscript foods-1887537 focused on the investigation of microwave and roasting process on pumpkin seed oil and full-fat mayonnaise quality produced from the extracted oil, respectively. Therefore, the topic of the manuscript is interesting and can provide useful information for the practice, as well. The manuscript is generally well written with a logic structure. Introduction section summarizes well the relevance of the study. Applied analytical methods (tocopherol, sterol HMF analysis; OIT, FTIR, mayonnaise characterization methods) are adequate and described clearly. The manuscript contains interesting and valuable results that are discussed with relevant references. Tables and figures represent well the experimental results.

Comments, suggestion:

What temperature was achieved during microwave irradiation (section 2.2, line 127-130)?

Please compare the energy need/efficiency/economy of microwave and roasting method.

Establishments in line 291-294 need reference.

Please improve the visibility of Figure 2 (mainly labels).

Round 2

Reviewer 1 Report

The authors have improved the manuscript as per my comments. The manuscript can be accepted for publication. 

Author Response

All suggestions have been reviewed point by point by the authors.

Thank you for your considerable contribution in improving the quality of the manuscript.

Reviewer 2 Report

The authors properly addressed my comments and improved the quality of the manuscript.

Author Response

Authors thank the Reviewer for her/his efforts in improving the quality of the manuscript.